# Burnout Syndrome in Physicians—Psychological Assessment and Biomarker Research

**DOI:** 10.3390/medicina55050209

**Published:** 2019-05-24

**Authors:** Tanya Deneva, Youri Ianakiev, Donka Keskinova

**Affiliations:** 1Department of Clinical Laboratory, Medical University, University Hospital “St. George”, 4002 Plovdiv, Bulgaria; 2Department of Psychology, University of Plovdiv Paisii Hilendarski, 4000 Plovdiv, Bulgaria; youri.ianakiev@gmail.com; 3Department of Applied and Institutional Sociology, University of Plovdiv Paisii Hilendarski, 4000 Plovdiv, Bulgaria; donka.keskinova@gmail.com

**Keywords:** burnout, physicians, stress biomarkers, health professionals, work-related stress

## Abstract

*Background and objectives*: Burnout is a syndrome typically occurring in work environments with continuous and chronic stress. Physicians are at increased risk for burnout, as a result of 24-h work, delayed work–life balance gratification, and the challenges associated with patient care. The aim of the present study was to evaluate the psychological parameters of burnout symptoms in relation to biomarkers of stress among physicians with different medical specialties. *Materials and methods*: A total of 303 physicians were contacted as potential participants. A comparison group of 111 individuals working outside medicine was used as a control to verify the results. The physicians were specialists in internal medicine, general surgery, pathology, and primary care. Serum cortisol, salivary cortisol, adrenocorticotropic hormone (ACTH), insulin (IRI), and prolactin levels were analyzed by chemiluminescence enzyme immunoassay (Access 2, Beckman Coulter). Fasting glucose in serum and glycated hemoglobin (HbA1C) in whole blood were measured using the automatic analyzer AU 480 Beckman Coulter system. Symptoms of burnout were measured with the Maslach Burnout Inventory (MBI). *Results*: The group with burnout presented significantly higher levels of serum and saliva cortisol, ACTH, prolactin, fasting glucose, and HbA1C compared with the control group. The correlation analysis between biomarkers showed a positive correlation with moderate strength between serum and saliva cortisol (*r* = 0.516, *p* = 0.01),as well as serum and saliva cortisol with ACTH (*r* = 0.418; *r* = 0.412, *p* = 0.01) and HbA1C (*r* = 0.382; *r* = 0.395, *p* = 0.01). A weak positive correlation was found between serum and saliva cortisol with prolactin (*r* = 0.236; *r* = 0.267, *p* < 0.01) and glucose (*r* = 0.271; *r* = 0.297, *p* < 0.01). In the multiple logistic regression model, saliva cortisol, HbA1C, and age were significantly associated with burnout (chi-square = 16.848, *p* < 0.032). *Conclusion*: Our findings demonstrated the interest of exploring biomarkers of stress related to burnout in health professionals.

## 1. Introduction

Burnout is a state of physical and emotional exhaustion, depersonalization, and a decreased sense of personal accomplishment caused by work-related stress. It is an outcome of chronic depletion of the individual’s coping resources resulting from prolonged exposure to stress, particularly work-related stress [1]. A prolonged response to chronic stressors of an emotional and interpersonal nature requires strong emotional involvement with the people who are the subjects of the work. The consequences are emotional (depersonalization with anger, frustration, demotivation, feeling of incompetence, and professional dissatisfaction),physical (pains, insomnia, depression, and illnesses), social (absenteeism and isolation), and behavioral (eating disorders, substance abuse, and workaholism) [2]. The relationship between health professionals and burnout syndrome is already well-known, as are the social, psychological, health, and work implications [3,4,5]. In the case of medical occupations, this might be manifested in a loss of empathy, “labeling” patients, and treating them as another “medical care.” Studies carried out among health professionals suggest that burnout results from prolonged exposure to stress factors. Some of the most important stress factors that have been reported include working long hours, frequently changing settings, being under 50 years of age, working weekends, managing highly demanding clinical interventions, inherent demands and stress of patient care, long and unsociable shift patterns, and an overall highly stressful environment. There are a few recent theoretical bases that explain the relationship between stress as a psychological factor and exhaustion. One study indicates that emotional exhaustion and health complaints usually emerge as indicators of the stress [6]. Other authors point out that emotional exhaustion is not only associated with work-related factors, but also with off-job physical activity and sleeping [7]. Several studies in the medical area have analyzed the relationship between sociodemographic, occupational, and personality variables and the occurrence of burnout syndrome. They have demonstrated that burnout is a global problem in emergency or critical care areas, oncology services, and primary care [8,9,10]. Healthcare professionals may develop symptoms such as anxiety, irritability, mood swings, insomnia, depression, and a sense of failure as a consequence of burnout [11,12,13,14,15]. These symptoms may ultimately lead to decreased job performance and poor patient care. On the other hand, adaptation to increased demands is regulated by the hypothalamic–pituitary–adrenal (HPA) axis, which helps the body maintain homeostasis during a stressful situation [16]. Since burnout is generally the result of a prolonged period of stress, it is often hypothesized that the HPA axis, a part of the neuroendocrine system involved in the regulation of reaction to stress, may be disturbed in individuals suffering from burnout [17,18]. Chronic exposure to stressors can contribute to permanent HPA axis activation [19]. As the major output of the HPA axis is the stress hormone cortisol, cortisol levels are considered to be different among subjects with burnout when compared to healthy people [20]. Since burnout is associated with chronic stress in the work environment, the levels of this syndrome could be related to daily cortisol secretion in health professionals [21]. It has been reported that hyperactivity of the HPA axis during a stress-induced situation may change to hypoactivity after long-term exposure to stressful circumstances [16,22]. Most studies on the behavior of this system focused on acute stress. During acute stress, the sympathetic part of the autonomous nervous system (ANS) and the HPA are activated. These are reflected in peripheral blood by release of catecholamines via the ANS and release of cortisol via the HPA axis. There is an increase in both heart rate and blood pressure. The immune system is temporarily suppressed, and metabolism becomes catabolic. Among people with burnout, increased incidences of flu-like illness, physical fatigue, irritability, back pain, and gastrointestinal problems have been reported [23]. Nevertheless, it is unlikely that these biological changes are responsible for the other symptoms of burnout (e.g., feelings of emotional exhaustion, detachment from work, and diminished competence). Therefore, several studies have investigated different biomarkers involved in the HPA, ANS, immune system, metabolic processes, antioxidant defense, and hormones (cortisol in saliva and blood, cholesterol, C-reactive protein, ACTH, prolactin, fibrinogen, etc.) for associations with symptoms of burnout. However, the results are conflicting [18,24,25,26,27]. Furthermore, the results indicate that no potential biomarkers for burnout were found, largely due to the incompatibility of studies.

Due to inconsistent findings from literary searches on potential stress biomarkers in occupational burnout, different types of biomarkers were selected for this study to investigate their association with burnout. The aim of the present study was to investigate the relationship between serum levels of cortisol, adrenocorticotropic hormone (ACTH), prolactin, insulin (IRI), glucose, salivary cortisol, glycated hemoglobin (HbA1C) in whole blood, and burnout syndrome in physicians with different medical specialties. The main study hypothesis was that burnout in physicians is associated with significantly higher values of the stress biomarkers investigated.

## 2. Materials and Methods

### 2.1. Participants

We invited 600 people to participate in the study, 186 of whom were excluded because they did not meet inclusion criteria (*n* = 112) or declined to participate (*n* = 74). Four hundred and fourteen people agreed to participate, and all their data were analyzed. A total of 303 physicians employed at the University Hospital St. George (Plovdiv, Bulgaria) were surveyed. They were selected personally and in a consecutive manner by a researcher after contact with all of the professionals working on the units. The physicians were specialists in internal medicine, general surgery, pathology, and primary care. A comparison group of 111 individuals working outside medicine was used as a control to verify the results. The control group participants were randomly selected when they were interviewed during routine laboratory investigations and met the criteria for inclusion in the survey after completing key demographic questionnaires. 

The exclusion criteria were as follows: Age under 18 years; anamnesis and clinical data for primary axis disorders; psychiatric disorders or using psychotropic drugs (antidepressants, sedatives, or hypnotics); inflammatory or immune diseases, cardiovascular disease, diabetes, rheumatic, peripheral blood disease, cancer, stroke, or metabolic or endocrinological abnormalities; BMI > 30 kg/m^2^; alcohol or drug abuse; and excessive smoking or caffeine.

All participants were given a document about the objectives and procedures of the study. The study was completely anonymous, and participants gave written informed consent. The study was conducted according to the principles of the Declaration of Helsinki, as revised in 2013,and was reviewed and approved by the Ethical Commission of the University of Plovdiv (P-244-1/22/02/2018).

### 2.2. Instruments

Symptoms of burnout were measured with the Maslach Burnout Inventory (MBI). This instrument is currently the most commonly used for evaluating burnout in healthcare professionals. The MBI consists of 22 elements, with scores based on the frequency of feelings related to the construction of exhaustion (Maslach and Jackson, 1981). The MBI’s three subscales were analyzed separately: Emotional exhaustion, depersonalization, and personal accomplishment. Mean values were calculated and subscales were categorized into “low,” “moderate,” and “high” degrees of burnout using the cut-off values suggested by Maslach [1]: For the subscale emotional exhaustion (EE), this translates into ≤18, 19–26, and ≥27 points, respectively; for the subscale depersonalization (DP), ≤5, 6–9, and ≥10 points, respectively; and for the subscale personal accomplishment (PA), ≤33, 39–34, and ≥40 points, respectively. Higher scores on the subscales emotional exhaustion and depersonalization indicate a higher degree of burnout, while a higher score on the subscale personal accomplishment indicates a lower degree of burnout.

Serum cortisol levels, salivary cortisol, plasma ACTH, IRI, and prolactin levels were analyzed by the chemiluminescence enzyme immunoassay method. Measurements were taken with an automatic device (Access 2, Beckman Coulter). Fasting glucose in serum and HbA1C in whole blood were measured by hexokinase assay and turbidimetric immunoinhibition methods, respectively, using the automatic analyzer AU 480 (Beckman Coulter system).

### 2.3. Procedures

All participants were invited to the study on a voluntary basis. First, before the survey, the purpose of the research was clarified. The participants were instructed on how to fill in the questionnaires and informed that the survey was anonymous and would not have any influence on their work or personal life. Second, sociodemographic information was obtained by a researcher in order to determine whether they met the inclusion criteria. Third, participants signed the written informed consent form and completed the self-administered evaluation and all the abovementioned questionnaires. Fourth, participants were given Salivette^®^ with instructions on how to collect saliva and conserve samples correctly by chewing a cotton swab for 60 s and placing it in the Salivette^®^. Venous blood and salivary samples were taken in the morning between 6 and 8 a.m., following the basic rules of specimen collection, and stored at −20 °C until the time of the analysis, but for no longer than two months according to the manufacturer’s instructions.

### 2.4. Data Analysis

Statistical calculations were carried out using IBM SPSS Statistics (v.25.0) and significance was fixed at *p* < 0.05. Quantitative variables were presented as mean and standard deviation (SD). Means were compared using a two-tailed *t*-test. Qualitative variables were presented as occurrence (*n*) and percentage (%). Column proportions were compared using a two-tailed *z*-tests. All tests were adjusted for all comparisons within a row of each innermost sub-table using the Bonferroni correction. Dependences between quantitative variables were checked using a Pearson correlation test. A multiple logistic regression model was made to identify predictive factors (independent variables) of burnout (dependent variables). A value of *p* < 0.05 for the Wald criterion was considered to denote regression coefficients significantly different from zero. The results are shown as odds ratios (Exp.) with 95% confidence intervals (95% CI) for EXP (B). The fit of the models was judged by the likelihood ratio test statistic.

## 3. Results

A description of the study population is detailed in Table 1. A total of 303 physicians were included in our study. The physicians were specialists in internal medicine (17.2%), general surgery (32.7%), and pathology (38.9%),as well as general practitioners (GP) (11.2%). Of these, 47.9% were women and 52.1% men. The average age was 48.6 years (SD = 9.9), with an age range of 23–65 years, and clinical work experience mean 19.3 years (SD = 9.8). Forty-seven percent (*n* = 143) of the physicians worked nightshifts under emergency conditions. The control group used (*n* = 111) was matched to the physicians by age, sex, and work experience.

A diagnosis of burnout (yes/no) was assigned if respondents presented high levels in at least two subscales (either EE and/or DP, associated or not with low PA) or in three subscales based on the following scores: EE >27, DP >10, and PA <33. According to this, 39.3% of the physicians (*n* = 119) and all of the controls (*n* = 111) did not show symptoms of burnout. In the EE subscale, 28.4% of the physicians (*n* = 86) without burnout showed a high score, and 15.2% of physicians (*n* = 46) indicated a higher degree of burnout in the three subscales—high levels of emotional exhaustion and depersonalization with a low level of personal accomplishment. The distribution of physicians by specialties in terms of diagnosis of burnout is shown in a Table 2.

To assess the overall extent of burnout symptoms, we analyzed the percentage distribution in the three dimensions of the burnout in the group of physicians by specialties. In terms of emotional exhaustion, 60.7% of doctors showed a high score. One hundred of the surgeons had a high score in the emotional exhaustion scale, followed by the groups of internists (80.8%) and GPs (61.8%). Regarding depersonalization, 26.4% of the group reported a high score, with the highest percentage again among surgeons. As for personal accomplishment, the results were high in 76.9%, with the largest share among pathologists (Table 3).

To identify parameters associated with a high degree of burnout, multivariate analyses were performed. A higher proportion of physicians working the nightshift under emergency conditions showed high scores on subscales of emotional exhaustion (74.8%) and depersonalization (38.5%) compared with those working full time (*p* = 0.000). Participants aged over 45 years also showed significantly higher emotional exhaustion compared with those under 45 years of age. Gender affected only emotional exhaustion. More often, men (67.7%) were associated with high values of this symptom than women (53.1%). In multivariate analyses, however, inpatient work was associated with higher emotional exhaustion (93.4%) and depersonalization (43.7%) compared with outpatient-based physicians. Lower personal accomplishment was found among primary care physicians (Table 4).

### Biochemical Parameters in Burnout Subjects

In this study, in order to assess potential biochemical variables associated with burnout, we analyzed serum and saliva cortisol, ACTH, prolactin, IRI, glucose, and HbA1C between the two groups of subjects—non-burnout (controls and physicians) and burnout physicians. The burnout group presented significantly higher levels for serum and saliva cortisol, ACTH, prolactin, fasting glucose, and HbA1C. We found that some of the physicians (*n* = 86) who did not have burnout, but showed a high score on the subscale of emotional exhaustion, also demonstrated significantly higher values for most of these biomarkers compared with the control group (Table 5).

The correlation analysis between biomarkers showed a positive correlation of moderate strength between serum and saliva cortisol (*r* = 0.516, *p* = 0.01), serum and saliva cortisol with ACTH and HbA1C, and fasting glucose levels and HbA1C. A weak positive correlation was found between serum and saliva cortisol with glucose and prolactin levels (Table 6). Established correlations also matched in the control group.

Multivariate logistic regression analysis was performed to identify predictive factors (independent variables) of burnout (dependent variable). From all possible predictive factors—biomarkers and demographics (gender, age, and work experience)—in the logistic regression model, we excluded work experience due to the high correlation with age (*r* = 0.80, *p* = 0.000) and IRI due to the lack of statistically significant difference between the values of this biomarker in the control and the doctors with burnout (*t* = 0.224, *p* = 0.823) (Table 7). The logistic regression model was statistically significant (chi-square = 24.217, *p* = 0.002). The model explained 91.0% (Nagelkerke *R*^2^) of the variance in exhaustion and classified correctly 98.1% of the cases. With the increase of age, salivary cortisol, and HbA1C, the probability of exhaustion increased.

## 4. Discussion

The present study about burnout and its relationship with stress biomarkers among physicians with different medical specialties is the first large-scale survey in the country and the region. In the present data, 39% of the physicians did not show a high degree of burnout on any of the subscales, which was comparable to that among hospital physicians in Hamburg and Germany, as well as among European family doctors [28,29]. The achieved mean scores on the MBI subscales (emotional exhaustion: 46.8, depersonalization: 9.9, and personal accomplishment: 41.1) were higher than those detected among hospital physicians in Hamburg, Germany [29]. 

In the present study, a higher percentage of physicians (mainly surgeons, followed by a group of internists) exhibited high scores of EE and DP. In our opinion, the obtained results were connected with the place of employment and the nature of the job: 24-h work, nightshifts, emergency conditions, and inpatient-based physicians. Similar results were shown in a study among physiotherapists in Poland, where they observed the strongest emotional exhaustion and depersonalization among physical therapists employed in hospital unlike physiotherapists employed in health centers [30]. Sex differences in the manifestation of burnout have been reported for different occupational groups. In a meta-analysis by Purvanova et al., including studies covering a range of professions, men showed higher degrees of depersonalization, whereas women showed higher emotional exhaustion [31]. Purvanova and her colleagues proved that women report emotional exhaustion as the main symptom of burnout, whereas men tend to show increased levels of depersonalization as the main symptom of burnout. Unlike Parvanova’s study, our results showed a higher EE score in male rather than female physicians. No gender differences were found in the personal accomplishment dimension. This is probably due to the fact that women effectively implement coping strategies for dealing with stress in work settings. In multivariate analyses, we found that physicians who work in inpatient settings showed higher emotional exhaustion. This is in contrast to a meta-analysis of worldwide studies, in which outpatient work was associated with a higher degree of emotional exhaustion [32]. In another study from the United States, internal medicine hospitalists and outpatient general internists did not differ regarding emotional exhaustion and depersonalization [33]. We also detected that lower personal accomplishment was more common among outpatient general practitioners, unlike the study by Roberts et al., where lower personal accomplishment was reported among hospitalists [33]. 

We considered that the higher levels of personal achievement among physicians in our study were due to the fact that the study was conducted in university hospitals, where the component “intellectual stimulation at work” is a determinant for personal accomplishment—opportunities for continuous qualification, training of students, and research. Consistently, in a study of Swiss physicians, lower exposure to continuing education was associated with a lower degree of personal accomplishment and a higher risk for depersonalization [34]. 

In accordance with the purpose of the study, several biomarkers were tested for association with burnout. Our results showed that blood concentration of serum and saliva cortisol, ACTH, prolactin levels, fasting glucose, and HbA1C were significantly higher in burnout physicians compared with healthy controls. Literature searches show a lack of unequivocal conclusions about the association of occupational burnout with stress biomarkers. In a systematic review of 31 studies on 38 biomarkers and meta-analyses, the authors concluded that there are no potential biomarkers in burnout due largely to the incompatibility of the studies stemming from the differences in the methods used to characterize patients and controls to assess biomarkers and to control for confounders [35]. Furthermore, in the literature, there are few studies related to biomarkers of stress among medical professionals. However, the fact that in our study individuals with burnout had significantly higher levels of serum and saliva cortisol than those without burnout confirmed previous findings concerning the cortisol–burnout relationship. De Vente et al. reported that increased levels of cortisol occurred among individuals with burnout but not in healthy controls [18]. Grossi et al. also found higher salivary cortisol levels in participants with burnout than in healthy individuals [36]. A recent study among health professionals working in a palliative care unit to evaluate the association between burnout dimensions and salivary secretion of cortisol showed that the release of cortisol in a one-dimension burnout group was higher than that in the control group for cortisol response upon waking and at bedtime [37]. According to this, we also found higher levels of salivary cortisol in the group of doctors with burnout symptoms in one-dimension:Emotional exhaustion. Similarly, we found significant differences in the levels of prolactin, ACTH, glucose, and HbA1C between controls and the group of doctors without burnout but who had high scores in emotional exhaustion. To our knowledge, the study of Grossi et al. [27] proved that high burnout in women is associated with high levels of HbA1C. 

The present data in our study on the significant positive correlation between ACTH, cortisol levels, IRI, HbA1C, and glucose support the idea of generally hyperactive HPA axis regulation after stressor response. However, studies focused on the relationship between burnout syndrome and the hyper/hypoactivation reaction process of the HPA axis showed inconsistent results [12,16,22,38,39]. 

Our results from regression the model to identify predictive factors of burnout showed that age, saliva cortisol, and blood concentration of HbA1C are significantly associated with burnout in physicians, and the predictive value of these biomarkers had the highest statistical significance in the scale of emotional exhaustion among burnout dimensions. Thus, our data confirmed the significant predictiveness of HbA1C in burnout found in the study by Metlaine et al. [40]. Today, salivary cortisol is routinely used as a biomarker of psychological stress [41]. The relationship between burnout and age has been proved in a number of studies [42,43,44]. Considering this, our study is the first of its kind in our country which reports that burnout in physicians, and particularly emotional exhaustion, is associated with HbA1C, saliva cortisol, and age. However, this study has some limitations which have to be taken into consideration when interpreting the results. Our study was descriptive, cross-sectional research. Therefore, it was not possible to analyze the causal relationships between variables. Future studies should be prospective and focused on physicians or a large group of health professionals to confirm the link between burnout and stress biomarkers.

## 5. Conclusions

In conclusion, our results indicate that blood concentration of serum and saliva cortisol, ACTH, prolactin levels, fasting glucose, and HbA1C are significantly higher in burnout physicians compared with healthy controls, which corroborates this study’s hypothesis. The significant positive correlation between ACTH, cortisol levels, IRI, HbA1C, and glucose support the relationship between burnout and HPA axis activation. Our results focus on the predictive role of biomarkers, in particular, saliva cortisol and HbA1C, in burnout syndrome. The present data confirm that there are some psychological and physiological aspects related to stress in the medical profession. Indeed, they may be relevant for further research in order to implement prevention programs aimed at reducing the negative aspects of professional distress. 

## Figures and Tables

**Table 1 medicina-55-00209-t001:** Characteristics of the study population.

Variable	Physicians	Controls	*p* Value for Comparisons
Female % (*n*)	47.9 (145)	53.2 (59)	*p* = 0.34
Male % (*n*)	52.1 (158)	46.8 (52)	
Age, mean (SD; range)	48.6 (9.9; 23–65)	46.6 (10.2; 25–64)	*p* = 0.07
GP % (*n*)	11.2 (34)		
Internists % (*n*)	17.2 (52)		
Pathologists % (*n*)	38.9 (118)		
Surgeons % (*n*)	32.7 (99)		
Length of work experience (years), mean (SD; range)	19.3 (9.8; 4–40)	18.6 (5.3; 7–37)	*p* = 0.39
Works full time % (*n*)	52.8 (160)		
Working on a nightshift % (*n*)	47.2 (143)		
Works in an inpatient setting % (*n*)	89.0 (269)		

**Table 2 medicina-55-00209-t002:** Burnout in the group of physicians and controls.

Burnout	GP (A)	Internists (B)	Pathologists (C)	Surgeons (D)	All Physicians (E)	Controls (F)
No	38.2% (13)	19.2% (10)	81.4% (96) ^AB^	-	39.3% (119)	100% (111) ^E^
No, but EE is high	-	51.9% (27) ^C^	9.3% (11)	48.5% (48) ^C^	28.4% (86)	-
Yes—EX and DP or EX and PA	52.9% (18) ^BCD^	9.6% (5)	9.3% (11)	18.2% (18)	17.2% (52)	-
Yes—EX, DP, and PA	8.8% (3)	19.2% (10)	-	33.3% (33) ^A^	15.2% (46)	-
Total	100% (34)	100% (52)	100% (118)	100% (99)	100% (303)	100% (111)

Cells contain percentage (number of cases). Compare column proportions with *z*-tests. For each significant pair, the key of the category with the smaller column proportion appears in the category with the larger column proportion. Significance level for upper case letters (A,B,C,D, and E): 0.05. If the column proportion is equal to zero or one, it was not used in comparisons.

**Table 3 medicina-55-00209-t003:** Extent of the degree of burnout symptoms in physicians and controls.

MBI		GP (A)	Internists (B)	Pathologist (C)	Surgeons (D)	All Physicians (E)	Controls (F)
Emotional exhaustion (EE)	low (<18)	-	-	38.1% (45)	-	14.9% (45)	61.3 (68) ^E^
	moderate (19–26)	38.2% (21)	19.2% (10)	43.2% (51)^B^	-	24.4% (74)	38.7 (43) ^E^
	high (>27)	61.8% (21) ^C^	80.8% (42) ^C^	18.6% (22)	100% (99)	60.7% (184)	-
Depersonalization (DP)	low (<5)	55.9% (19)	61.5% (32)	69.5% (82)^D^	48.5% (48)	59.7% (181)	92.8 (103) ^E^
	moderate (6–9)	35.3% (12)^B^	9.6% (5)	21.2% (25)	-	13.9% (42)	7.2 (8) ^E^
	high (>10)	8.8% (3)	28.8% (15) ^C^	9.3% (11)	51.5% (51) ^ABC^	26.4% (80)	-
Personal accomplishment (PA)	low (<33)	61.8% (21) ^BD^	19.2% (10)	-	33.3% (33)	21.1% (64)	-
	moderate (34–39)	17.6% (6)	-	-	-	2.0% (6)	22.5 (25) ^E^
	high (>40)	20.6% (7)	80.8% (42) ^A^	100% (118)	66.7% (66) ^A^	76.9% (233)	77.5 (86) ^E^

Cells contain percentage (number of cases).Compare column proportions with *z*-tests. For each significant pair, the key of the category with the smaller column proportion appears in the category with the larger column proportion. Significance level for upper case letters (A,B,C,D,E, and F): 0.05. If the column proportion is equal to zero or one, it was not used in comparisons.

**Table 4 medicina-55-00209-t004:** Burnout symptoms in physicians.

MBI	Indicator	Sex		Age		Type of Diagnostic		Work Time		
		Women	Men	≤45	>45	Outpatient	Inpatient	Full Time	Nightshift	
Emotional exhaustion >27	% (*n*)	53.1% (77)	67.7% (107)	52.6% (62)	69.8% (122)	61.8% (21)	93.4% (141)	48.1% (77)	74.8% (107)	
	*z*-test	*p* < 0.01		*p* < 0.01		*p* < 0.001		*p* < 0.001		
Depersonalization >10	% (*n*)	22.8% (33)	29.7% (47)	28.3% (28)	25.5% (52)	8.8% (3)	43.7% (66)	15.6% (25)	38.5% (55)	
	*z*-test	ns		ns		*p* < 0.001		*p* < 0.001		
Personal accomplishment <33	% (*n*)	17.9% (26)	24.1% (38)	24.2% (24)	19.6% (40)	61.8% (21)	28.5% (43)	19.4% (31)	23.1% (33)	
	*z*-test	ns		ns		*p* < 0.001		ns		

Compare column proportions with *z*-tests, ns = not significant (*p* ≥ 0.05).

**Table 5 medicina-55-00209-t005:** Biomarkers in burnout and control subjects.

Biomarkers	Physicians and Control, no Burnout (*n* = 230) (A)	Physicians, no Burnout, but EE is High (*n* = 86) (B)	Physicians, with Burnout (*n* = 98) (C)
Serum cortisol	277.1 (98.2)	331.4 (112.0) ^A^	310.6 (100.3) ^A^
Saliva cortisol	25.0 (11.7)	33.3 (9.8) ^A^	33.2 (9.7) ^A^
ACTH	14.7(4.7)	16.4 (5.4)	17.0 (5.7) ^A^
Prolactin	210.1 (126.6)	256.4 (116.3) ^A^	255.1 (140.3) ^A^
Insulin (IRI)	6.7 (3.4)	6.4 (3.6)	6.4 (3.3)
HbA1C	5.0 (0.3)	5.1 (0.4)	5.2 (0.4) ^A^
Glucose	5.6 (0.5)	5.7 (0.6) ^A^	5.8 (0.6) ^A^

Cells contain mean (SD). Compare column means with *t*-tests. For each significant pair, the key of the category with the smaller column proportion appears in the category with the larger column proportion. Significance level for upper case letters (A,B, and C): 0.05.

**Table 6 medicina-55-00209-t006:** Pearson correlation between biomarkers in physicians with burnout (*n* = 98).

Biomarkers	Serum Cortisol	Saliva Cortisol	ACTH	Prolactin	Insulin (IRI)	HbA1C	Glucose
Serum cortisol	1.000	0.516 **	0.418 **	0.236 **	0.168	0.382 **	0.271 **
Saliva cortisol	0.516 **	1.000	0.412 **	0.267 **	0.146	0.395 **	0.297 **
ACTH	0.418 **	0.412 **	1.000	0.033	0.099	0.134	0.012
Prolactin	0.236 **	0.267 **	0.033	1.000	0.135	0.161	0.142 *
Insulin (IRI)	0.168	0.146	0.099	0.135	1.000	0.093	0.256 *
HbA1C	0.382 **	0.395 **	0.134	0.161	0.093	1.000	0.468 **
Glucose	0.271	0.297	0.012	0.142*	0.256 **	0.468 **	1.000

** Correlation is significant at the 0.01 level (two-tailed); * Correlation is significant at the 0.05 level (two-tailed).

**Table 7 medicina-55-00209-t007:** Model of multiple logistic regression.

Factors	B	S.E.	Wald	df	P	OR	95% CI	
							Lower	Upper
Age	0.262	0.084	9.877	1	0.002 *	1.300	1.104	1.531
Sex	−1.576	1.229	1.644	1	0.200	0.207	0.019	2.301
Serum cortisol	−0.001	0.010	0.019	1	0.889	0.999	0.978	1.019
Saliva cortisol	0.474	0.137	11.903	1	0.001 *	1.606	1.227	2.103
ACTH	−0.245	0.118	4.288	1	0.083	0.783	0.621	0.987
Prolactin	0.001	0.003	0.152	1	0.697	1.001	0.995	1.008
HbA1C	4.728	2.500	3.577	1	0.039 *	113.117	1.842	15196.953
Glucose	−0.176	1.185	0.022	1	0.882	0.839	0.082	8.549

* Significance level (*p* ≤ 0.05).

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
