# Peer review of "Burnout Syndrome in Physicians—Psychological Assessment and Biomarker Research"

_medicina, 2019, doi:10.3390/medicina55050209_

Round 1

Reviewer 1 Report

The study presents a relevant and current theme, especially at the international level, due to the individual and organizational repercussions of burnout. However, to contribute to the advancement of knowledge, it is necessary to adjust the weaknesses indicated by section:

In the introduction, one must strengthen the knowledge gap, align with the objective, and include the hypotheses of the study.

In the Material and Methods, the description of the steps should be reviewed, with a view to reproducibility of the study. The description can be guided by an international guideline, such as STROBE.

In the results, the homogeneity of the study group was not presented in relation to the control group, attested by statistical tests. In relation to table 2, Maslach and Leiter have recently indicated five latent MBI profiles and suggest other cutoff points to determine them (https://www.sciencedirect.com/science/article/pii/S2213058615300188). Considering the objectives, it was not clear why the authors inserted a multiple model, mainly logistic regression, since the Pearson correlation was used in the univariate analysis. In addition, there are variables in the model that are theoretically collinear (multicollinear).

The conclusion must meet the goal. To review the statement "In conclusion, our findings demonstrated the interest of biomarkers, in particular saliva cortisol and HbA1C levels in the characterization of burnout in physicians who are more prone to burnout prevalence." because in order to sustain it, it would require a study of accuracy (positive predictive value, negative predictive value, sensitivity and specificity).

Author Response

Thank you very much for reviewing our manuscript. We also greatly appreciate the complimentary recommendations. We revised the manuscript accordingly yours comments and suggestions. The followings are our point-by-point responses:

1. In the introduction, one must strengthen the knowledge gap, align with the objective, and include the hypotheses of the study.

Response: Following the prescriptions we reviewed and enriched the introduction by including a hypothesis corresponding to the objectives of the study.

2. In the Material and Methods, the description of the steps should be reviewed, with a view to reproducibility of the study. The description can be guided by an international guideline, such as STROBE.

Response: We reviewed and corrected the section on "Material and Methods" to improve the reproducibility of the study. We included the corresponding sections: Participants, instruments, procedures, data analysis and we supplemented missed detail in this section

3. In the results, the homogeneity of the study group was not presented in relation to the control group, attested by statistical tests. In relation to table 2, Maslach and Leiter have recently indicated five latent MBI profiles and suggest other cutoff points to determine them (https://www.sciencedirect.com/science/article/pii/S2213058615300188).

Response: Maslach and Leiter (2016) proposed a new approach to understanding the burnout experience through five latent burnout profiles. In our study, we follow the indications of Maslach's certified test which propose three dimensions of chronic stress in a professional environment where the diagnosis of burnout is presented as high levels of stress in two or three dimensions. In this case the main focus of our research is related to the correlation between the results of the Burnout syndrome assessment of Maslach and the results of the stress biomarkers.  Another study devoted entirely to the latent profiles of burnout and biomarkers could be considered.

4. Considering the objectives, it was not clear why the authors inserted a multiple model, mainly logistic regression, since the Pearson correlation was used in the univariate analysis. In addition, there are variables in the model that are theoretically collinear (multicollinear).

Response: Logistic regression was used to determine the effects of age, gender, and biochemistry on the likelihood of participants having Burnout. Correlation only establishes a connection. Theoretically, they can be collinear but empirically they are not (in Table 6 there is no coefficient higher than 0.7)

5. The conclusion must meet the goal. To review the statement "In conclusion, our findings demonstrated the interest of biomarkers, in particular saliva cortisol and HbA1C levels in the characterization of burnout in physicians who are more prone to burnout prevalence." because in order to sustain it, it would require a study of accuracy (positive predictive value, negative predictive value, sensitivity and specificity).

Response: We accept the recommendation and we have revised the conclusion in accordance with the purpose and the finding results of the study.

Reviewer 2 Report

The presented work analyzes a topic of interest and a fundamental importance in the performance of the health profession. Currently there are works that analyze exhaustion and stress, but also those that analyze the relationship of both with stress biomarkers. So the presented work needs big changes to be improved and susceptible of publication:

1. The introduction must be improved. There are few theoretical bases that explain the aspects to analyze, as well as the theory on which the authors are based, as well as the possible relationships between stress (psychological factor) and exhaustion. There are studies that analyze, recent studies in this same magazine platform, for example:

https://www.mdpi.com/search?q=burnout&authors=&journal=&article_type=&search=Search

2. They should include hypotheses in the introduction.

3. The Material and methods section should include the corresponding sections: Participants, instruments, procedures, data analysis.

4. The MBI consists of 22 elements, with scores based on the frequency of feelings related to the construction of exhaustion (Maslach and Jackson, 1981).

5. The analyzes of the data should be reviewed, in some cases, they have not been indicated correctly in the tables. For example, table 1,% is not included, so it can be confused with the average score. The table is quite confusing. The same happens with table 2.

6. Uses nonparametric tests and does not appear in the control group, except in Table 5, where they are compared with other groups without indicating the type of test. Finally a regression is performed, but it is not indicated.

7. I think we should establish an analysis of the effects of stress variables on exhaustion, as well as thinking about a model in which the mediating variables between stress and exhaustion can be indicated.

8. The conclusion is minimal, the analysis must be improved to obtain a more relevant conclusion.

Author Response

Thank you very much for reviewing our manuscript. We also greatly appreciate the complimentary recommendations. We revised the manuscript accordingly yours comments and suggestions. The followings are our point-by-point responses:

1. The introduction must be improved. There are few theoretical bases that explain the aspects to analyze, as well as the theory on which the authors are based, as well as the possible relationships between stress (psychological factor) and exhaustion. There are studies that analyze, recent studies in this same magazine platform, for example:

https://www.mdpi.com/search?q=burnout&authors=&journal=&article_type=&search=Search

Response: Following the prescriptions we have enriched the introduction.

2. They should include hypotheses in the introduction.

Response: Following the prescriptions we have included a hypothesis corresponding to the objectives of the study. The Material and methods section should include the corresponding sections: Participants, instruments, procedures, data analysis.

3. The Material and methods section should include the corresponding sections: Participants, instruments, procedures, data analysis.

Response: We reviewed and corrected the section on "Material and Methods" to improve the reproducibility of the study and we included the recommended sections: participants, instruments, procedures, data analysis in the text of the revised manuscript.

4. The MBI consists of 22 elements, with scores based on the frequency of feelings related to the construction of exhaustion (Maslach and Jackson, 1981).

Response: As suggested, we have replaced the sentence in the manuscript (p.4, line 132-133).

5. The analyzes of the data should be reviewed, in some cases, they have not been indicated correctly in the tables. For example, table 1,% is not included, so it can be confused with the average score. The table is quite confusing. The same happens with table 2.

Response: We agreed with the comments and we have corrected the tables 1, 2, 3, 4 and 5 in the revised manuscript

6. Uses nonparametric tests and does not appear in the control group, except in Table 5, where they are compared with other groups without indicating the type of test. Finally a regression is performed, but it is not indicated.

Response: After the statistical processing of the results, it turned out that some physicians (n=119) did not show any symptoms of Burnout and practically did not differ from the control group in terms of the average values of the biomarkers studied. That is why we have included them in the control group in Table 5 when comparing the average values of these indicators to the participants with Burnout. However, we have made some adjustments under Table 5 for more accurate clarity of the information. We have indicated in Table 7 “multiple logistic regression”.

7. I think we should establish an analysis of the effects of stress variables on exhaustion, as well as thinking about a model in which the mediating variables between stress and exhaustion can be indicated.

Response: In the present study we evaluated psychological parameters of burnout in association with biomarkers of stress. The main focus of our research is related to the correlation between the results of the Burnout syndrome assessment of Maslach and the results of stress biomarkers. Another study devoted entirely to the model in which the mediating variables between stress and exhaustion could be considered.

8. The conclusion is minimal, the analysis must be improved to obtain a more relevant conclusion.

Response: We accept the recommendation and we have revised the conclusion in accordance with the purpose and the finding results of the study.

Reviewer 3 Report

This manuscript examines physicians burnout symptoms and evaluates psychological parameters of burnout in association with biomarkers of stress. In general, the manuscript is interesting and captures an important topic in burnout research, however, several language mistakes should be corrected, and more details should be provided concerning the selection of the variables and control group demographics. I have mentioned few of the language mistakes but the authors should read through the manuscript carefully and correct the rest of them. Below are my detailed comments and suggestions for the authors.

Introduction

1.    Pp.2 line 49: add ‘stress factors’ in the sentence ‘Some of the most important have been reported..’

2.    Pp. 2 line 65: add ‘are’ in the sentence ‘During acute stress… and HPA are activated’

3.    Pp. 2 line 72: replace ‘but’ with ‘however,’

4.    A better rationale should be provide on why serum levels of cortisol, ACTH, prolactin, insulin (IRI), glucose, salivary cortisol, 76 glycated hemoglobin (HbA1C) in whole blood were selected for this study (e.g., due to the conflicting findings of previous studies different kind of biomarkers were examined?)

5.    Please read carefully through the manuscript and correct all the language mistakes.

Materials and methods

6.    Pp. 2 line 78: remove ‘and’

7.    How the comparison group was selected for the study?

8.    Did all the participants who were contacted agree to participate in the study?

9.    Please give the information at Table 1 already here.

10.  Please provide the demographics also for the control group.

11.  Pp. 3 line 113: replace ‘it is’ with ‘The study was’

Results

12.  Please include the control group in Table 2 and Table 3

13.  Shouldn’t control group be separate from the physicians in the Table 5?

14.  Table 7 should state ‘multiple logistic regression’

Discussion

15.  Pp. 7 line 210: replace ‘in female ones’ with ‘among female physicians’

16.  Burnout might also manifest as altered experiences of one of the burnout dimensions, however, in the present study burnout was conceptualized as a combination of two or more dimensions which might have affected the results. Please discuss further.

Author Response

Thank you very much for reviewing our manuscript. We also greatly appreciate the complimentary recommendations. We revised the manuscript accordingly yours comments and suggestions. The followings are our point-by-point responses:

Introduction

1. Pp.2 line 49: add ‘stress factors’ in the sentence ‘Some of the most important have been reported.’

Response: As you have suggested, we have added ‘stress factors’ in the sentence ‘Some of the most important have been reported’ (Pp.2 line 54 in revised manuscript)

2. Pp. 2 line 65: add ‘are’ in the sentence ‘During acute stress… and HPA are activated’

Response: As you have suggested, we have added ‘‘are’ in the sentence ‘During acute stress… and HPA are activated’ (Pp.2 line 83 in revised manuscript)

3. Pp. 2 line 72: replace ‘but’ with ‘however,’

Response: As suggested, we replaced ‘but’ with ‘however in the sentence “Therefore several studies have investigated different biomarkers involved in the HPA, ANS, immune system, metabolic processes, antioxidant defence, hormones, (cortisol in saliva and blood, cholesterol, C-reactive protein, ACTH, prolactin, fibrinogen, and the like) for association with symptoms of burnout, however the results are conflicting (Pp.3 line 94 in revised manuscript)

4. A better rationale should be provide on why serum levels of cortisol, ACTH, prolactin, insulin (IRI), glucose, salivary cortisol, 76 glycated hemoglobin (HbA1C) in whole blood were selected for this study (e.g., due to the conflicting findings of previous studies different kind of biomarkers were examined?)

Response: We agreed with this comment and we have carried out a better rationale in the revised manuscript (Pp.3 line 97-104 in revised manuscript).

5. Please read carefully through the manuscript and correct all the language mistakes.

Response: As you have recommended we have corrected language mistakes.

Materials and methods

6. Pp. 2 line 78: remove ‘and’

Response: As you have recommended we have removed “and”, but we reviewed the section “Materials and methods” entirely according to the recommendations of the other reviewer. We included the recommended sections: participants, instruments, procedures, and data analysis

7. How the comparison group was selected for the study?

Response: The control group participants working outside medicine were randomly selected when they were interviewed in the routine laboratory investigations and met the criteria for inclusion in the survey after completing demographic key questionnaires. All participants signed a written informed consent to participate in the survey.

8. Did all the participants who were contacted agree to participate in the study?

Response: We invited 600 people to participate in the study, 186 of whom were excluded because they did not meet inclusion criteria (n=112) or declined to participate (n=74). 414 people agreed to participate and all their data was analysed.

9. Please give the information at Table 1 already here.

Response: the information at Table 1 is a part of the results of the study. That’s why it was proposed into the results section. However, we have supplemented the information on the survey participants in “Material and methods”, and we corrected  Table 1 with  the demographic characteristics of the control group.

10. Please provide the demographics also for the control group.

Response: we have added the demographic data of the control group in Table 1.

11. Pp. 3 line 113: replace ‘it is’ with ‘The study was’.

Response:  We have replaced ‘it is’ with ‘The study was’. (Pp.3 line 126 in revised manuscript).

Results

12. Please include the control group in Table 2 and Table 3

Response: We have included the control group in the Table 2 and Table 3

13. Shouldn’t control group be separate from the physicians in the Table 5?

Response: After the statistical processing of the results, it turned out that some physicians (n=119) did not show any symptoms of Burnout and practically did not differ from the control group in terms of the average values of the biomarkers studied. That is why we have included them in the control group in Table 5 when comparing the average values of these indicators to the participants with Burnout. However, we have made some adjustments under Table 5 for more accurate clarity of the information.

14. Table 7 should state ‘multiple logistic regression’

Response: We have made Table 7 ‘multiple logistic regression’.

Discussion

15. Pp. 7 line 210: replace ‘in female ones’ with ‘among female physicians’

Response: We have replaced in female ones’ with ‘among female physicians’ (Pp.11 line 334 in revised manuscript).

16. Burnout might also manifest as altered experiences of one of the burnout dimensions, however, in the present study burnout was conceptualized as a combination of two or more dimensions which might have affected the results. Please discuss further.

Response: In our study, we follow the indications of Maslach's certified test which proposes three dimensions of chronic stress in a professional environment where the diagnosis of burnout is presented as high levels of stress in two or three dimensions. In this case, the main focus of our research is related to the correlation between the results of the Burnout syndrome assessment of Maslach and the results of stress biomarkers.

Round 2

Reviewer 1 Report

After adjustments the text contributes to the advancement of knowledge and, therefore, justifies its publication.

Author Response

Thank you very much for reviewing our manuscript. Thank you for your constructive suggestions and approval.

Reviewer 2 Report

1. They must clearly indicate the data of the participants, it is not clear they are 303 or 111.

2. They must indicate the Bioethics Committee that has been submitted to the study. ESSENTIAL FOR PUBLICATION.

3. A study of this type should not be published, without analyzing the role of the variable. This should be an objective of the investigation, should not be analyzed superficially, for after another work, as the authors say, analyze it. This work is the moment to analyze in depth the variables that are presented. Otherwise, the statements that do not become a reality.

4. The interpretation of the data is incorrect. The model explains 92.9% (Nagelkerke R2 229) of the variance in exhaustion and correctly classifies 97.6% of the 230 cases. With increasing age, salivary cortisol and HbA1C, the probability of exhaustion increases. It must be reviewed properly.

5. The conclusions are briefly repeated the results.

Author Response

The followings are our point-by-point responses:

1. They must clearly indicate the data of the participants, it is not clear they are 303 or 111.

Response: The information concerning the number of participants was already clearly indicated in the section 2. Materials and methods: 2.1. Participants – line 99-106.

2. They must indicate the Bioethics Committee that has been submitted to the study. ESSENTIAL FOR PUBLICATION.

Response: The study was approved by Ethical commission of the University of Plovdiv (P-244-1 / 22/02/2018) and we indicated in the revised manuscript (Pp 3, line 115-116)

3. A study of this type should not be published, without analyzing the role of the variable. This should be an objective of the investigation, should not be analyzed superficially, for after another work, as the authors say, analyze it. This work is the moment to analyze in depth the variables that are presented. Otherwise, the statements that do not become a reality.

Response: As indicated, we analyzed the role of the variables. The multivariate logistic regression analysis was made to identify prediction factors (independent variables) of burnout (dependent variable). From all possible predictive factors - biomarkers and demographics (gender, age and work experience) in the logistic regression model, we excluded the work experience due to the high correlation with age (r = 0.80, p = 0.000) and the Insulin (IRI) due to the lack of statistically significant difference between the values of this biomarker in the control and the doctors with Burnout (t = 0.224, p = 0.823). We presented the new data in Table 7 and corrected the text as follow: (pp. 7 lines 228-238) 

4. The interpretation of the data is incorrect. The model explains 92.9% (Nagelkerke R2 229) of the variance in exhaustion and correctly classifies 97.6% of the 230 cases. With increasing age, salivary cortisol and HbA1C, the probability of exhaustion increases. It must be reviewed properly.

Response: Following the prescriptions we reviewed the interpretation of the data (Pp 7. Line 234-235)

5. The conclusions are briefly repeated the results.

Response: We have revised the conclusion in accordance with the purpose and the finding results of the study as suggested by two others reviewers.